# Nanocrystal superlattices as phonon-engineered solids and acoustic metamaterials

Nuri Yazdani[1], Maximilian Jansen [1], Deniz Bozyigit[1], Weyde M.M. Lin [1], Sebastian Volk[1], Olesya Yarema[1], Maksym Yarema[1], Fanni Juranyi [2], Sebastian D. Huber [3] & Vanessa Wood [1]

Phonon engineering of solids enables the creation of materials with tailored heat-transfer properties, controlled elastic and acoustic vibration propagation, and custom phonon–electron and phonon–photon interactions. These can be leveraged for energy transport, harvesting, or isolation applications and in the creation of novel phonon-based devices, including photoacoustic systems and phonon-communication networks. Here we introduce nanocrystal superlattices as a platform for phonon engineering. Using a combination of inelastic neutron scattering and modeling, we characterize superlattice-phonons in assemblies of colloidal nanocrystals and demonstrate that they can be systematically engineered by tailoring the constituent nanocrystals, their surfaces, and the topology of superlattice. This highlights that phonon engineering can be effectively carried out within nanocrystal-based devices to enhance functionality, and that solution processed nanocrystal assemblies hold promise not only as engineered electronic and optical materials, but also as functional metamaterials with phonon energy and length scales that are unreachable by traditional architectures.

---

[1] Materials and Device Engineering Group, Department of Information Technology and Electrical Engineering, ETH Zurich, Zurich CH-8092, Switzerland. [2] Laboratory for Neutron Scattering and Imaging, Paul Scherrer Institute, CH-5232 Villigen PSI, Switzerland. [3] Institute for Theoretical Physics, ETH Zurich, 8093 Zürich, Switzerland. Correspondence and requests for materials should be addressed to V.W. (email: vwood@ethz.ch)

Colloidal semiconducting or metallic nanocrystals (NCs) can be assembled from solution into densely packed 2D or 3D superlattices (Fig. 1)[1–5]. A key feature of these NC superlattices is their tunability[6]. The NC building blocks of the superlattice can be synthesized with precise control over composition, size, shape, and surface-terminating ligands[7,8]. This tunability has been used to engineer the electronic and optical structure of NCs, and is recognized as key for a wide array of applications, including LEDs, solar cells and photo-detectors, transistors, phase-change memory, and thermoelectric devices[9–13]. Furthermore, the inter-NC spacing and packing of the NCs into these superlattices can be tuned by the size and shape of the NCs and choice of ligand[1,14], with structures ranging from primary crystal structures (e.g., cubic[3,15], BCC[14], FCC[16], and hexagonal[2]) to complex binary systems (e.g., NaCl, MgZn2…)[5,17]. This multi-parameter tunability can potentially be exploited to control the collective vibrational structure of the NC superlattice, which would enable the design of new materials via phonon engineering.

The vibrational structure of NC superlattices is complex. The types of vibrations and phonons expected in NC-superlattices are shown in Fig. 1. A superlattice containing $N_{NC}$ NCs, each containing $N_A$ atoms in its core and ligand shell, has $3N_{NC} N_A$-6 degrees of atomic freedom. $N_{NC}(3N_A$-6) of these are the vibrations of the atoms making up the ligand and the NCs. A majority of the vibrational modes of organic ligand species, $g_{lig}(\omega)$ occur at high energies (i.e., ~100 meV and above). Vibrations of the atomic lattice of the NC cores, $g_{NC}(\omega)$, including the bulk-like phonons of the NC core and the localized vibrations of the surface atoms[18,19], occur at intermediate energies (~1–100 meV). The remaining $6N_{NC} - 6$ atomic degrees of freedom contribute to excitations of the NC superlattice, related to displacements ($3N_{NC} - 6$) and hindered rotations ($3N_{NC}$) of the NCs about their equilibrium positions. The $3N_{NC} - 6$ vibrational modes of the NC superlattice (i.e., superlattice phonons, with density $g_{SL}(\omega)$) can be long range and coherent, as evidenced previously by the observation of acoustic standing wave excitations in NC superlattice thin films[20].

Here, using the well-studied system of lead sulfide (PbS) NCs linked with dithiol ligands, we model and directly measure $g_{SL}(\omega)$ for 3D NC-superlattices. We show that $g_{SL}(\omega)$ can be systematically tuned through selection of the NC size and choice of ligand. This demonstrates that NC-superlattices are not only promising designer electronic and optical materials, but also solution-processable, phonon-engineerable solids. In particular, we highlight that the design of long-range phonons in NC superlattices can be used to achieve materials with novel low temperature properties, and that the vibrational structure in NC

superlattices enables them to be used as functional 2D and 3D acoustic metamaterials spanning energy and length scales not achievable using standard methods.

## Results

**Modeling superlattice phonons.** To gain an understanding of the expected phononic structure and energy scale for a NC superlattice, we model it as a three-dimensional mass-spring system, where the NCs are the masses, $m$, and the surface terminating ligands are the springs providing a force constant, $k$, between neighboring NCs.

In a mass-spring model, the energy of the phonon modes will scale with $\sqrt{k/m} = \sqrt{n(r)k_{lig}/m}$, where $n(r)$ is the number of ligands providing an interaction between neighboring nanocrystals of radius $r$, and $k_{lig}$ is the effective spring constant of a single ligand. In the case of dithiol ligands, we assume that longitudinal stretch/compression of the dithiol carbon backbone will dominate the overall mechanical interactions between neighboring NCs, and ignore the impact of shearing/flexing of the ligands. We determine $k_{Lig}$ using density functional theory (DFT) to compute the total energy $E(x)$ of the ligand as a function of its compression or extension, $x$, with $E(x) - E(0) = 1/2 k_{lig}x^2$. As an example, in Fig. 2a we show the calculated $E(x)$ for hexane-dithiol (HDT) (blue dots) and the fit (black line), which highlights that the carbon-backbone of the dithiol ligands exhibit a harmonic response. We compute $k_{lig}$ for four dithiol ligands of varying carbon-backbone lengths, ethanedithiol (EDT, $N_C = 2$), butane-dithiol (BDT, $N_C = 4$), hexanedithiol (HDT, $N_C = 6$), and dodecanedithiol (DDT, $N_C = 12$), and we find a decreasing $k_{lig}$ with increasing carbon backbone length (Fig. 2b). $n(r)$ will depend on the structure of the NC superlattice, but will scale with $r^\alpha$, with $2 \geq \alpha \geq 1$, the upper and lower bounds corresponding to superlattice structures where neighboring NCs are face-sharing or edge sharing respectively.

Small angle x-ray scattering studies of PbS NC thin films have demonstrated their assembly into FCC, BCC, and related structures[14,21,22], depending on the NC size and ligand, with facet to facet separations in the superlattice corresponding to the nominal length of the ligands employed. The resulting superlattice lattice constants are therefore $a_{SL} \approx 2r_{NC} + \ell_{lig}$, where $r_{NC}$ and $\ell_{lig}$ are the NC radius and nominal ligand length respectively. For the thiol ligands studied here, $\ell_{EDT} = 0.45$ nm, $\ell_{BDT} = 0.69$ nm, $\ell_{HDT} = 0.94$ nm, $\ell_{DDT} = 1.7$ nm. For PbS NC superlattices with thiol ligands a FCC structure is expected for smaller NCs (<3 nm radius) (Fig. 2c)[16].

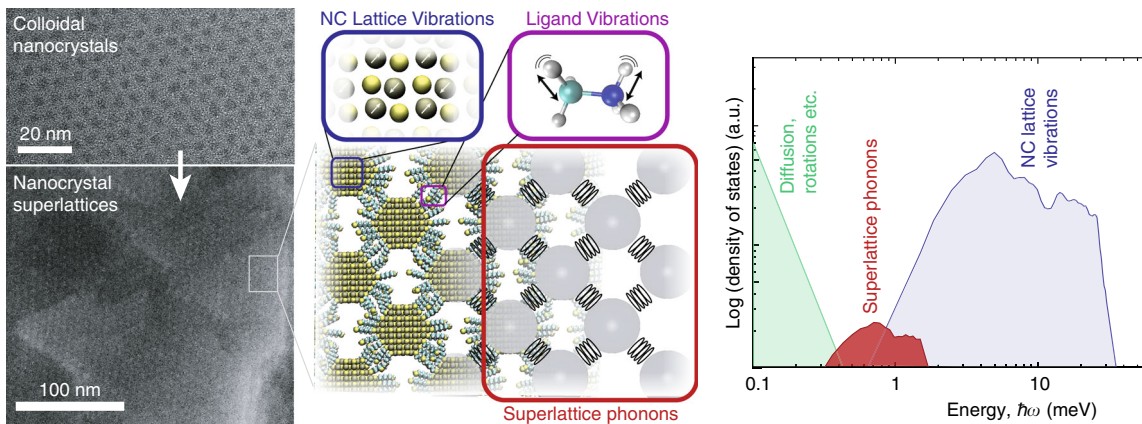

**Fig. 1** Phonons in Nanocrystal Superlattices. Schematic of classes of vibrations and phonons in nanocrystal superlattices, along with an illustrative plot depicting their density of states for a PbS nanocrystal superlattice with dithiol ligands

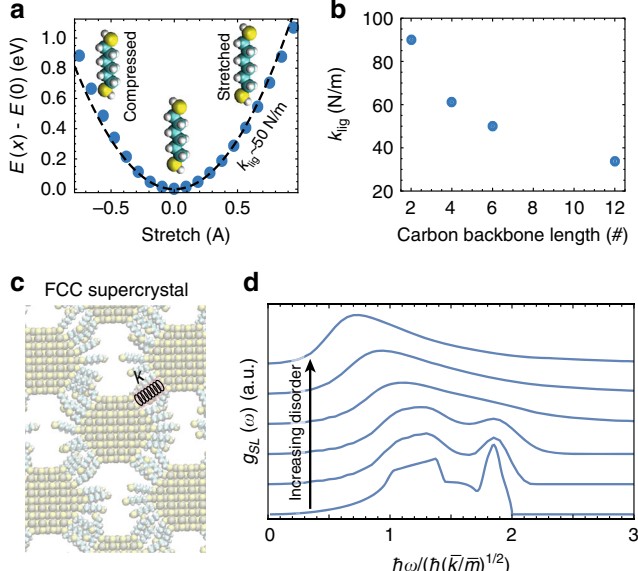

**Fig. 2** Modeling the Superlattice Phonon Density of States **a** The total energy $E$ of an HDT ligand plotted versus the stretch of the ligand, $x$, shows the characteristic parabolic dependence of a spring. **b** Computed effective spring constants of dithiol ligands. **c** Illustration of PbS NC-solids depicting a [100] cross-section through a FCC superlattice. **d** The expected $g_{SL}(\omega)$ for an FCC lattice with no disorder (bottom line), and with increasing disorder in the NC masses ($\sigma_m = 10, 20, 50, 100, 200\%$), and a uniform distribution of spring constants, $k \in (0.5, 1.5)$ N/m, i.e., $\beta = 1$

The analytical form for the phonon density of states for an FCC lattice is shown in Fig. 2d. However, a real NC superlattice will have some disorder that will depend on the quality of the NC synthesis and the methods and conditions used to assemble the NCs into the superlattice[1,3]. We can numerically compute $g_{SL}(\omega)$ for disordered superlattices by constructing the dynamical matrix, **D**, for the mass-spring model and diagonalizing it, $\mathbf{D}\boldsymbol{\varphi} = -\omega^2\boldsymbol{\varphi}$, where $\omega$ is the frequency of the normal mode $\boldsymbol{\varphi}$. We take the masses $m_i$ from a normal distribution with standard deviation $\sigma_m$ and assume a uniform distribution of spring constants $k_{ij}$, for the springs connecting mass $i$ and $j$, with width $\beta$. With increasing disorder, the sharp transverse-acoustic (TA) and longitudinal-acoustic (LA) peaks of $g_{SL}(\omega)$ broaden and eventually merge into a single broad, low energy peak (analogous to the Boson Peak of disordered atomic/molecular lattices)[23,24], at high amounts of disorder. For solution processed PbS superlattices, which will be inherently disordered, we can expect a $g_{SL}(\omega)$ between the two extremes in Fig. 2d.

Our modeling provides easy insight into how $g_{SL}(\omega)$ of NC-superlattices can be tailored by changing the mass of the NCs (e.g., by changing their size or composition) and/or the effective spring constants (e.g., though the packing geometry or the type of ligand). For FCC PbS NC superlattices fabricated with typical NC sizes ($r = 1.5–4$ nm) and thiol ligands, we can expect TA/LA Van Hove singularities (peaks in the $g_{SL}(\omega)$) at energies $\hbar\omega_{TA} \sim 1.2\hbar\sqrt{k/m}$, $\hbar\omega_{LA} \sim 2\hbar\sqrt{k/m}$, within a range of ~0.4–2 meV (Supplementary Note 1).

**Measuring superlattice phonons**. To measure these superlattice vibrations, we turn to inelastic neutron scattering (INS)[25]. Both inelastic x-ray scattering (IXS)[19] and INS have been employed to measure the vibrational density of states stemming from atomic vibrations within the NCs ($g_{NC}(\omega)$). While

IXS has the advantage that measurements can be performed on thin films (i.e., tens of nanometers), it does not have sufficient energy resolution to probe $g_{SL}(\omega)$. We therefore turn to INS, performed at the time-of-flight spectrometer FOCUS at SINQ, and use an incident neutron energy of 3.27 eV (wavelength of 5.0 Å) to obtain an energy resolution of ~82 μeV (FWHM) at 0 meV energy transfer. Samples consist of ~8–10 g of PbS NCs synthesized using an upscaled approach[26] and condensed into a powder of NC superlattice crystallites with a solution-based ligand exchange (Supplementary Note 2)[25]. While the large sample sizes required for INS prevents us from employing state-of-the-art superlattice formation procedures[1–4], our calculations indicate that superlattice phonons will still be observed in a disordered system (Fig. 2d). Extraction of $g_{SL}(\omega)$ from INS is non-trivial due to the overlap with quasi-elastic scattering centered at 0 energy transfer, which results from reorientation motion of the surface terminating ligands[27]. We therefore develop a fully automated analysis procedure to extract $g_{SL}(\omega)$ from INS and apply it identically to all measurements on all samples (Supplementary Note 3).

In Fig. 3a, we plot the extracted NC superlattice phonon density of states, $g_{SL}(\omega)$, for a series of NC-superlattices fabricated with increasing NC radii of $r = 1.6, 1.9, 2.5,$ and $3.3$ nm, all with HDT ligands. The extracted $g_{SL}(\omega)$ for all samples consists of a broad double peaked distribution, characteristic of the phonon density of states of a disordered FCC superlattice, and at an energy range consistent with our modeling above. The position of the first peak (TA) of the distribution shows the expected energy scaling with NC radius of $\hbar\omega \propto r^{-1}$ (Fig. 3b). In Fig. 3c, we plot the $g_{SL}(\omega)$ for a series of four samples with 1.6 nm radii NCs fabricated with different ligands. As expected, the energetic scale of $g_{SL}(\omega)$ increases with a decrease in the carbon backbone length of the dithiol ligand. While the scaling for the BDT, HDT, and DDT follows the trend $k_{lig}^{1/2}$ (Fig. 3d), the characteristic energy of the EDT sample is lower than expected from the mass-spring model. As we discuss in Supplementary Note 1, this likely stems from interactions between the superlattice phonons and the lattice vibrations of the NC cores, reducing the energies of the superlattice phonons.

These experimental results validate that phonons of NC-superlattices can be modeled as mass-spring systems and systematically engineered. Due to their solution processability as thin films or packed powder samples, they offer phonon engineering from the sub-micron to centimeter length scales and at energies in the 0–2 meV range, opening up new form factors, energy bandwidths, and use cases for meta-materials.

**Low-Temperature properties of superlattice phonons**. Due to the low energy ($\hbar\omega/k_B \sim 10–20$ K) and low number density (~$10^3–10^4$ fewer than the atomistic vibrations of the NC and the ligands), the impact of superlattice phonons on material properties will be particularly evident at ultra-low temperatures where the thermal occupation of ligands and NC core vibrations is negligible[28,29]. Interestingly, even though the superlattice phonons energies are low, their corresponding group velocities are similar to those in bulk PbS due to the several-nm length scale lattice constant of the NC-superlattice $a$ (e.g., for 1.6 nm NCs with BDT $v_{g,LA} \propto \omega a/\pi$ ~1.5 meV 3.9 nm/$\hbar\pi$ ~3 $10^3$ m/s, whereas for bulk, $v_{g,LA} \propto$ ~10.5 meV 0.6 nm/$\hbar\pi$ ~3 $10^3$ m/s). Therefore, below 4 K, properties such as the lattice heat capacity and thermal conductivity of the NC superlattice can match or even slightly exceed that of the bulk. This is emphasized in Fig. 4a, which shows the lattice thermal conductivity, $\kappa_{lat}$, from the longitudinal acoustic (LA) phonon

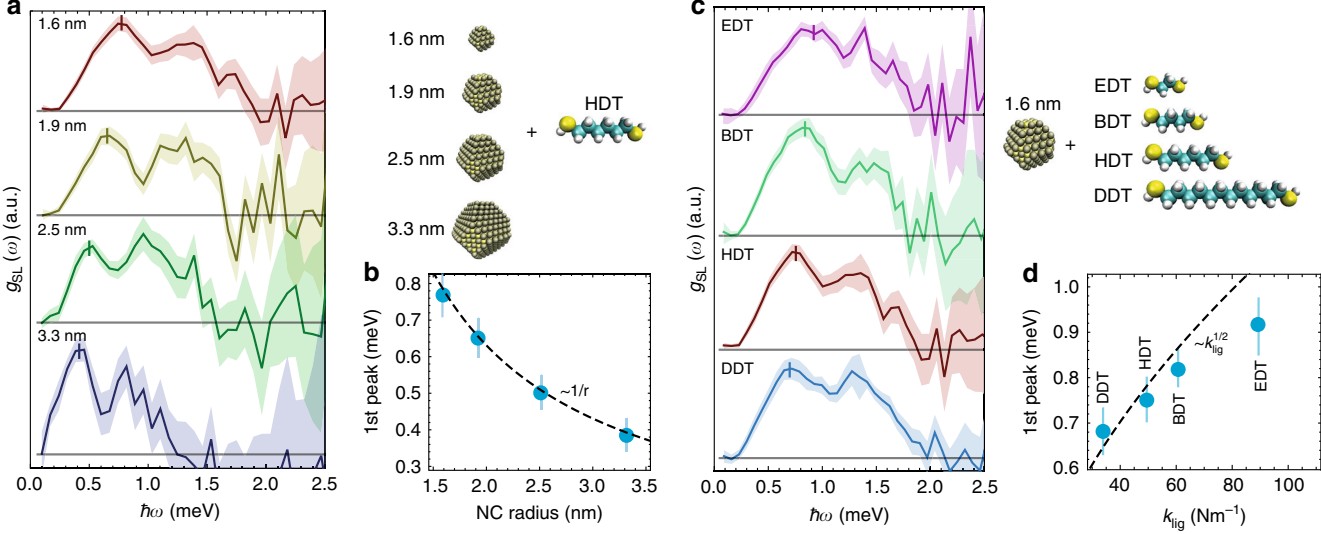

**Fig. 3** Engineering Nanocrystal Superlattice Phonons.: **a** Extracted $g_{SL}(\omega)$ for NC-solids fabricated with NCs of varying size (1.6 nm, 1.9 nm, 2.5 nm, and 3.3 nm), the shaded regions indicating the error, measured at 300 K. The energies of $g_{SL}(\omega)$ are found to scale with $r^{-1}$ **b**, taken from the position of the first peak of the $g_{SL}(\omega)$ indicated by the vertical ticks, determined through a fitting of a normal distribution to $g_{SL}(\omega)$ from $\hbar\omega = 0$ to ~0.1 meV above the fitted peak position for each samples, errorbars correspond to the standard error of the fit peak position. **c** Extracted $g_{SL}(\omega)$ for NC-solids fabricated with 1.6 nm NCs with EDT, BDT, HDT, or DDT ligands, measured at 300 K. **d** A weak scaling of the energies of $g_{SL}(\omega)$ are found, taken from the scaling of the position of the first peak of $g_{SL}(\omega)$, whose position and error were determined using the same approach as those presented in **b**

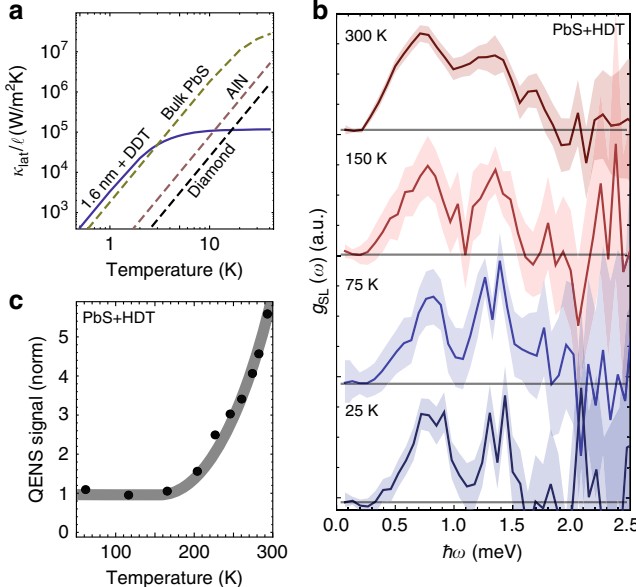

**Fig. 4** Temperature Dependence of Nanocrystal Phonons.: **a** Temperature dependence of the lattice thermal conductivity normalized by phonon mean-free-path-length of 1.6 nm PbS NC + DDT superlattice, as well as bulk PbS, diamond, and aluminum-nitride. **b** Extracted $g_{SL}(\omega)$ at 300 K, 150 K, 75 K, and 25 K for a NC superlattice fabricated with 1.6 nm NCs and HDT ligands. **c** Quasi-Elastic-Neutron-Scattering (QENS) intensity integrated from 20 to 50 μeV for the 1.6 nm NCs and HDT ligand superlattice as a function of temperature

mode normalized by the phonon mean free path:

$$\kappa_{\text{lat}}/\ell = \frac{4 k_B}{\pi^2} \frac{v_g}{a^3} \left(\frac{k_B T}{\hbar\omega_{LA}}\right)^3 \int_0^{\hbar\omega_{LA}/k_B T} \frac{x^4 e^x}{(e^x - 1)^2}\, dx. \quad (1)$$

Since carrier mobility in NC superlattice is temperature activated[30], NC-superlattices will be robust insulators at low-temperature. Figure 4a shows that, at low temperatures, the thermal conductivity of the PbS NC superlattice could be two orders of magnitude larger than the well-known, electrically insulating thermal conductors, diamond and aluminum-nitride. This assumes of course highly ordered superlattices with large mean free paths, but large mean free paths in these materials should be in general be facilitated by the large $a$.

A prerequisite to the NC superlattice being used as a phonon engineered material at low temperatures is of course reliant on the phononic structure being maintained at these temperatures. We therefore measure the $g_{SL}(\omega)$ of the $r = 1.6$ nm NCs and HDT ligand sample between 25 and 300 K (Fig. 4b). The extracted $g_{SL}(\omega)$ is similar for all temperatures with a sharpening and slight blueshift of the features with decreasing temperature, which is consistent with increased ordering (leading to lower disorder in $k$) and a densification of NC-solid at low temperature (leading to an increase in $k$)[16]. We propose that this ordering and densification comes from suppression of the reorientational motion of the ligands at low temperatures. To test this hypothesis, we perform fixed-window scans on the PbS-HDT sample at the neutron backscattering spectrometer MARS at SINQ, which provides a high energy resolution (~13 μeV) and enables the rapid acquisition of the temperature evolution of the reorientational motion of the ligands[27]. The QENS intensity is constant below ~175 K (Fig. 4c), and then starts to rapidly increase, indicating an increased flexibility of the ligands. These findings highlight that NC superlattices maintain their phononic structure across a broad temperature range, and that cooling to cryogenic temperatures can reduce the extent of disorder in the NC superlattice and inter-NC interactions. These results suggest that NC superlattices will be highly effective electronically insulating, thermal conductors at ultra-low temperatures, which could make them of interest, for example, as a thermal dissipation coating in cryo-CMOS devices.

## Discussion

In summary, this demonstration of phonon engineering in NC superlattices indicate that they are poised to become not only designer electronic and optical materials[2,3], but also phononic solids, where the phonon density of states ($g_{SL}(\omega)$, $g_{NC}(\omega)$, and $g_{lig}(\omega)$) can be engineered through the choice of ligand and the composition, size, shape, and surface of the NCs[18,31].

While here we have focused on acoustic phonons of simple monatomic superlattices, this platform can be extended towards the creation of complex phononic structures traditionally targeted by the meta-materials community[32–34]. Phonon engineering has been predominately demonstrated with acoustic metamaterials[35–39] on a macroscopic scale[40,41]. The design of specialized organic[42] and inorganic[43] ligands can enable more complex inter-NC mechanical interactions, beyond the stretch/compressive forces provided by the thiol ligands investigated here. Including the full range of elastic interactions, such as sheer and torsional forces, would unleash the power of complex phonon design algorithms[44]. Furthermore, by controlling the shape, and therefore exposed surfaces of the NCs, the functional group of the ligands can be tailored to link between specific NC facets[7,8,45], enabling the definition of specific directional interactions in the superlattice. Although assembly techniques for NC-superlattices must continue to be developed so that they can compete with macroscopic metamaterials in terms of complexity, NC-superlattices enable phonon engineering at length and energy scales not easily achievable with existing platforms.

Moreover, the ability to independently tune phononic and electronic properties in NC-superlattices is particularly promising. Phonon engineering at the atomistic level is typically limited by the fact that electronic and phononic properties are intrinsically linked. In the case of NC superlattices, electronic properties (like phononic properties) are tunable by the choice of NC size[46,47], shape[48], chemical composition[49,50], and ligand[51]. However, electronic coupling between neighboring NCs and the reorganizational energy associated charge transfer govern electronic properties, while the mechanical properties of the ligand and the mass of the nanocrystal dominate the long range superlattice vibrations. Thus, electronic properties could be changed while phononic properties remain the same by, for example, placing a shell around the NC cores that changes the NC bandgap and electronic coupling between neighboring nanocrystals but keeps the NC masses the same. Alternatively, switching the ligand (e.g., from ethanedithiol to benzenedithiol) could change the effective spring constant of the system while keeping the electronic properties the same (i.e., inter-NC spacing and electronic coupling would be similar). Furthermore, the ability to tune the spatial confinement of the charge carriers (e.g., with multilayer shells, alloys, and electronegativity of ligands), to select optical band gap (e.g., with NC size), and to introduce phononic bandgaps (e.g., with complex superlattice structures using NCs of different masses), provides opportunities to selectively tune electron-phonon, phonon–photon, and phonon–phonon interactions in NC superlattices.

The ease of their solution fabrication, the new length scales and energies achievable with NC superlattices, and their unique low temperature properties will hopefully stimulate new dialog about the opportunities for these systems as phononic materials and devices.

## Methods

**Density functional theory calculations of thiol spring constants**. Geometry optimization, electronic structure calculations, and *ab initio* molecular dynamics (AIMD) were performed within the CP2K program suite utilizing the quickstep module[52]. Calculations are carried out using a dual basis of localized Gaussians and plane-waves[53], with a 300 Ry plane-wave cutoff. Double-Zeta-Valence-Polarization (DZVP)[54], Goedecker–Teter–Hutter pseudopotentials[55] for core electrons, and the Perdew–Burke–Ernzerhof (PBE) exchange correlation functional are used for all

calculations. Convergence to $10^{-8}$ in Self-Consistent Field calculations is always enforced.

The equilibrium geometry (stretch $x = 0$) of the thiols are first determined through a full relaxation of the molecules. For each stretch amount, $x$, the molecules are uniformly stretched such that the sulfur-sulfur distance is stretched by $x$, the geometry of the molecule is then fully relaxed with only the position of the two sulfurs fixed. The resulting energies of the minimized geometries are then taken as $E(x)$.

**Calculating $g_{SL}(\omega)$**. We construct the dynamical matrix as described in the main text, and diagonalize it to obtain $g_{SL}(\omega)$. With disorder, we average $g_{SL}(\omega)$ over 100 stochastic realizations of the dynamical matrix. Calculations are performed on lattices with dimensions of $16 \times 16 \times 16$ conventional BCC/FCC unit cells with periodic boundary conditions.

**Sample preparation**. PbS NC solutions were synthesized using an upscaled synthesis approach[26]. For the 1.6 nm NCs, 2 separate syntheses were mixed into a single solution for preparation of the ligand series. PbS NC-solid powders were prepared identically to our previous INS experiments[25], using a fixed molar concentration of 8 mM for all four dithiol ligands.

**INS/QENS measurements**. Each measured sample consisted of ~8 g of PbS powder, prepared as described above. The powder is held in a cylinder (d = 12 mm, h = 40–50 mm) made of 3 layers of aluminum foil. Data reduction for both INS and QENS measurements were performed using the DAVE software package[56].

INS was performed at the cold-neutron time-of-flight spectrometer (FOCUS) at Swiss Spallation Neutron Source SINQ. For INS, a Vanadium standard with dimensions 5.0 cm × 1.5 cm × 0.2 mm thick was used to measure the incident neutron energy and for detector efficiency calibration, and determination of the instrument resolution. For INS spectra all samples were measured for ~8 h. Measurement of the empty sample holder subtracted as background.

QENS experiments were performed at the indirect time-of-flight backscattering spectrometer MARS at SINQ. Scattered neutrons with energy of 1.85 meV (6.65 Å) were filtered with the (006) reflection of the mica analyzers in near-backscattering configuration. The choice of the incident energy range defined the energy transfer window. For the QENS signal intensity, the scattering was summed between 0.02 and 0.05 meV, which represents well the quasielastic part of the spectrum. The temperature dependence of the background is negligible; therefore, it was not measured, nor subtracted from the data.

## Data availability

All relevant data are available from the authors.

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

## Acknowledgements

We acknowledge support from the Swiss National Science Foundation through the Project #175889, the Quantum Sciences and Technology NCCR, and an ETH Research Grant #42-12-2. Computations were supported by a grant from the Swiss National Supercomputing Centre (CSCS; project ID s831). This work is based on experiments performed at the Swiss spallation neutron source SINQ, Paul Scherrer Institute, Villigen, Switzerland.

## Author contributions

N.Y., D. B and V.W. devised the work, N.Y. performed the modeling, N.Y. and M.J. performed the calculations, M.Y. and O.Y. synthesized materials, N.Y., D.B., S.V. and W.M.M.L. prepared and characterized the INS/QENS samples, N.Y., D.B. and F.J. performed the neutron scattering experiments, N.Y. analyzed results, N.Y., V.W., S.H. and F.J. wrote the manuscript with input from all other authors.

## Additional information

**Competing interests:** The authors declare no competing interests.



