## [Peer Review File · Nature Communications]

Reviewers' Comments:

Reviewer #1:

Remarks to the Author:

The authors use inelastic neutron scattering to probe the phonon density of states of a superlattice made of PbS nanocrystals. They probe different samples, containing nanocrystals of different sizes and different type (length) of ligands showing that the vibrational properties of this lattice of nanocrystals can be tuned to a certain degree.

The manuscript is very well written, concise and the topic is timely. The weakness of the manuscript is the relative lack of new/unexpected/exciting science and gain of knowledge. I am not able to judge the difficulty, novelty and the quality of the INS measurements. Nevertheless, I am convinced that the work will attract a lot of attention and be used as stepping stone for the next generation of experiments that will unveil interesting physical effects such as the ones described as outlook and in the introduction and conclusions, and that have to do with the coupling to electronic degrees of freedom. I therefore suggest to publish this manuscript in Nature Comm. Specific comments and questions.

1) Is the superlattice 2D or 3D?

2) The reason for the 6N-6 vibrations (instead of 3N-6) was not clear to me

3) "are also be tunable" should be corrected

4) The meaning of "are bright as well", was not clear to me.

5) The reason for the red shift of the Gsl with disorder is not described. Maybe a brief comment about this would be beneficial.

Reviewer #2:

Remarks to the Author:

The authors present a convincing picture for the observation and tunability of vibrational modes of nanocrystal superlattices and provide arguments for possible functionality of these assemblies as acoustic metamaterials. The extensive synthetic and neutron spectroscopy effort across the size and ligand size make the conclusions significant although the original individual signal is weak. The novelty and possible applications of the work make this paper of interest for publication and the paper is overall enjoyable to read. However, the manuscript -in its present form- fails to answer important questions concerning the data analysis and interpretation, as outlined below. Further, there are a few minor style issues that can be easily addressed.

1. There is no detailed discussion of the structure of the assemblies or of the lattice constant. It is particularly important to provide an experimental estimation of the typical interparticle distance (which could be obtained easily by electron microscopy or small-angle x-ray or neutron scattering).

2. In the discussion in line 56-62, the author should specify that shearing/flexing of the ligand is neglected. Although the results seem to justify that this is valid this is a simplification that needs to be stated.

3. The reference to the appearance of a Boson peak requires more precision. In the cited references to glass physics, configuration/structural disorder rather than force constant disorder appears as the crucial parameter.

4. In the discussion around line 129, the authors express the phonon group velocity as a function of a lattice constant. This is approximately correct but no lattice constant is discussed in the manuscript (beyond 'several nm' – this can be clarified, see point 1). As a starting point, the group velocity should be first defined as $v_g = dw/dk$, which in the long-wavelength limit is $\sim w/k$. A quick estimation based on the provided data (energy for TA and LA Van Hove singularities and nanocrystallite size) indicates that the group velocity is rather close to the speed of sound in PbS.

This claim thus needs to be better supported. Note also that Eq. 3 in the supplementary information provides means to obtain a parametric form for the group velocities.

5. The interesting but bold claim in line 137 needs further discussion or better context, as it implicitly assumes that the mean free path can be comparable to diamond or AlN. How likely is this in nanocrystallite assemblies?

6. Significance of the scattering data. The authors have made a great effort in the automatic data analysis, part of which is present in Fig. S3. However, a direct visual of the scattering function in linear scale in the 0.5 – 3 meV range is important and will make the existence of the peaks e.g. in Fig. 4B more clear.

7. Figure S3D suggest a weakness in the modelling above 4 meV, which is unlikely to affect conclusions but needs to be addressed. In the Supplementary Material, the authors state that the TA Van Hove singularity is at 6 meV. This is inconsistent with the core phonon term being shown as linear in $g(\omega)/\omega$.

Minor points:

- Line 26: NC is not defined.
- Line 35: body-centered tetragonal and rhombohedral assemblies of cubic/cuboctahedron particles have been observed as well, and a phase diagram has been derived as a function of shape and truncation (see Wetterskog et al., *Nanoscale*, 2016, DOI: 10.1039/C6NR03776C and references therein). Where are the assemblies herein poised in such a phase diagram?
- Line 41: most, but not all vibrational modes of the organic ligands are at higher energies.
- Line 80: the use of the indices ij,k , is not understandable here without further definition.
- Line 87: TA/LA peak energies should be referred to also as Van Hove singularity.
- Line 95: 3.27 meV.
- Line 150: QENS rather than QUENS.
- Line 152-154: revise grammar. 'do' hurts the readability and the subject of 'reduce' is unclear.
- Line 368: specify the temperature for each measurement in the caption.
- Line 382: the information of Fig. 4C is insufficiently discussed in the main manuscript.
- For SI Fig. 2: specify what type of absorption spectroscopy was used and with what experimental setup.

Raphael Hermann

Reviewer #3:

Remarks to the Author:

The manuscript looks at super-lattices made of PbS nanocrystal colloids connected together through various ligands. The mechanical properties of these structures are tuned by changing the number of atoms in the colloids (hence their mass) and by varying the length of the ligands (hence their stiffness). The manuscript argues that these super-lattices can be used as functional acoustic-metamaterials. The manuscript has several shortcomings that need to be adequately addressed before I can recommend publication in *Nature Communications*.

1) The current version of the manuscript makes it very hard to understand what is the significance of this work in the context of metamaterials. Very little evidence is given of what kind of functional metamaterials could be engineered with this technique. For instance (and this is just an example), a hot topic of research currently is how to implement topological metamaterials that support unidirectional surface modes (Kozin et al, *Phys Rev B* 98, 125115, 2018). This functionality relies on the existence of Dirac cones in the phononic crystal (Torrent and Sanchez-Dehesa, *Phys Rev Lett* 108, 174301, 2012). Can the NC super-lattices be designed to have this desirable feature? Another example alluded to in the introduction is isolation. There are several methods to achieve

acoustic isolation with passive (Mei et al, Nat Comms 3, 756, 2019) and active (Popa et al, Nat Comms 9, 5299, 2018) metamaterials. Where would the structure presented in the manuscript fit in the space of existing methods?

2) How is the NC lattice different from what has been done before given that structures made of PbS NCs connected through various ligands have been reported before (see [8]). Essentially, what exactly is the novelty brought by this work?

3) The manuscript shows that various components of the lattice are tunable (e.g. Fig. 2), but what is the space of effective stiffness and NC mass that is available in these structures, and how accurately can we design for a given stiffness or NC colloid mass?

4) Figs. 2D, 3A, and 3B are hard to interpret without numbers on vertical axis. Is the g_{SL} peak the only interesting feature in these plots? Also, it would be interesting to know how the expected and measured g_{SL} compare.

5) The stiffness shown in the caption of Fig. 2 has no units. The term NC is never defined.

We thank the reviewers for their careful inspection of the paper and their insightful comments. We have revised the manuscript accordingly, and believe it improved considerably as a result. Below, we reply point-by-point to the reviewers' questions, comments, and suggestions.

Reviewer #1 (Remarks to the Author):

The authors use inelastic neutron scattering to probe the phonon density of states of a superlattice made of PbS nanocrystals. They probe different samples, containing nanocrystals of different sizes and different type (length) of ligands showing that the vibrational properties of this lattice of nanocrystals can be tuned to a certain degree. The manuscript is very well written, concise and the topic is timely. The weakness of the manuscript is the relative lack of new/unexpected/exciting science and gain of knowledge. I am not able to judge the difficulty, novelty and the quality of the INS measurements. Nevertheless, I am convinced that the work will attract a lot of attention and be used as stepping stone for the next generation of experiments that will unveil interesting physical effects such as the ones described as outlook and in the introduction and conclusions, and that have to do with the coupling to electronic degrees of freedom.. I therefore suggest to publish this manuscript in Nature Comm. Specific comments and questions.

1) Is the superlattice 2D or 3D?

The superlattices could be 2D, but the ones explored here are 3D. We now state this explicitly: "Colloidal semiconducting or metallic nanocrystals (NCs) can be assembled from solution into densely packed 2D or 3D superlattices... Here, using the well-studied system of lead sulfide (PbS) NCs linked with dithiol ligands, we model and directly measure $g_{SL}(\omega)$ for 3D NC-superlattices." We have also added a paragraph outlining the structure of the PbS NC-superlattices (see also Reviewer 2, comment 1).

2) The reason for the $6N-6$ vibrations (instead of $3N-6$) was not clear to me

Thanks for this question. In total there are $3N_{\text{NC}}N_{\text{A}} - 6$ degrees of freedom. Each NC has $3N_{\text{A}} - 6$ vibrational modes of the NC core and ligands. The difference then, is $3N_{\text{NC}}N_{\text{A}} - 6 - N_{\text{NC}}(3N_{\text{A}} - 6) = 6N_{\text{NC}} - 6$.

We have updated the sentence to now read: "The remaining $6N_{\text{NC}} - 6$ atomic degrees of freedom contribute to excitations of the NC superlattice, related to displacements ($3N_{\text{NC}} - 6$) and hindered rotations ($3N_{\text{NC}}$) of the NCs about their equilibrium positions." We believe this now should make it clear for a reader that the vibrational degrees of freedom of the NC superlattice is indeed $3N_{\text{NC}} - 6$.

3) "are also be tunable" should be corrected

This has been corrected.

4) The meaning of "are bright as well", was not clear to me.

We have replaced "bright" with "promising". Considering that semiconductor NCs are used extensively for their optical emission, the use of "bright" can indeed be confusing.

5) The reason for the red shift of the Gsl with disorder is not described. Maybe a brief comment about this would be beneficial.

This point also coincides with point 3 of reviewer 2, and we provide a response below.

Reviewer #2 (Remarks to the Author):

The authors present a convincing picture for the observation and tunability of vibrational modes of nanocrystal superlattices and provide arguments for possible functionality of these assemblies as acoustic metamaterials. The extensive synthetic and neutron spectroscopy effort across the size and ligand size make the conclusions significant although the original individual signal is weak. The novelty and possible applications of the work make this paper of interest for publication and the paper is overall enjoyable to read. However, the manuscript -in its present form- fails to answer important questions concerning the data analysis and interpretation, as outlined below. Further, there are a few minor style issues that can be easily addressed.

1. There is no detailed discussion of the structure of the assemblies or of the lattice constant. It is particularly important to provide an experimental estimation of the typical interparticle distance (which could be obtained easily by electron microscopy or small-angle x-ray or neutron scattering).

We now provide a paragraph where we discuss the structure of the PbS NC superlattices, including the lattice constant approximation. There are a number of publications, which have focused on characterizing the structure of PbS NC-superlattices based on different preparation methods.

“Small angle x-ray scattering studies of PbS NC thin films have demonstrated their assembly into FCC, BCC, and related structures,^{9,21,22} depending on the NC size and ligand, with facet to facet separations in the superlattice corresponding to the nominal length of the ligands employed. The resulting superlattice lattice constants are therefore $a_{SL} \approx 2r_{NC} + \ell_{Lig}$, where r_{NC} and ℓ_{Lig} are the NC radius and nominal ligand length respectively. For the thiol ligands studied here, $\ell_{EDT} = 0.45$ nm, $\ell_{BDT} = 0.69$ nm, $\ell_{HDT} = 0.94$ nm, $\ell_{DDT} = 1.7$ nm. For PbS NC superlattices with thiol ligands a FCC structure is expected for smaller NCs (< 3nm radius) (**Figure 2C**).^{11”}

2. In the discussion in line 56-62, the author should specify that shearing/flexing of the ligand is neglected. Although the results seem to justify that this is valid this is a simplification that needs to be stated.

We have modified the following sentence to address this concern: “In the case of dithiol ligands, we assume that longitudinal stretch/compression of the dithiol carbon backbone will dominate the overall mechanical interactions between neighboring NCs, and ignore the impact of shearing/flexing of the ligands.”

3. The reference to the appearance of a Boson peak requires more precision. In the cited references to glass physics, configuration/structural disorder rather than force constant disorder appears as the crucial parameter.

Our aim, in the small section referenced (see also reviewer 1 point 5) is to give the reader an idea of what form of $g(w)$ can be expected for our NC-superlattices, and not to provide a thorough and systematic study of the impact of disorder on the vibrational properties of materials. The reference we provide by Taraskin discusses the emergence of a Boson peak with force constant disorder. We have reiterated at the end of the paragraph the objective of the calculations:

“With increasing disorder, the sharp transverse-acoustic (TA) and longitudinal-acoustic (LA) peaks of $g_{SL}(\omega)$ broaden and eventually merge into a single broad, low energy peak (analogous to the Boson Peak of disordered atomic/molecular lattices),^{29,30} at high amounts of disorder. For PbS NC superlattices studied here, which will be disordered due to their preparation through large scale solution processing, we can expect a $g_{SL}(\omega)$ between the two extremes of Figure 2D.”

4. In the discussion around line 129, the authors express the phonon group velocity as a function of a lattice constant. This is approximately correct but no lattice constant is discussed in the manuscript (beyond ‘several nm’ – this can be clarified, see point 1). As a starting point, the group velocity should be first defined as $v_g = d\omega/dk$, which in the long-wavelength limit is $\sim \omega/k$. A quick estimation based on the provided data (energy for TA and LA Van Hove singularities and nanocrystallite size) indicates that the group velocity is rather close to the speed of sound in PbS. This claim thus needs to be better supported. Note also that Eq. 3 in the supplementary information provides means to obtain a parametric form for the group velocities.

Indeed, we use a parametric expression for the group velocities, in which we use the long-wavelength limit, $v_g \equiv \omega a_{SL}/\pi$, where a_{SL} is the superlattice lattice constant. We have now clarified this paragraph, writing:

“Interestingly, even though the superlattice phonons energies are low, their corresponding group velocities are similar to those in bulk PbS due to the several-nm length scale lattice constant of the NC-superlattice a (e.g. for 1.6 nm NCs with BDT $v_{g,LA} \propto \omega a/\pi \sim 1.5 \text{ meV } 3.9 \text{ nm } /\pi \sim 3 \cdot 10^3 \text{ m/s}$, whereas for bulk, $v_{g,LA} \propto \sim 10.5 \text{ meV } 0.6 \text{ nm } /\pi \sim 3 \cdot 10^3 \text{ m/s}$).”

5. The interesting but bold claim in line 137 needs further discussion or better context, as it implicitly assumes that the mean free path can be comparable to diamond or AlN. How likely is this in nanocrystallite assemblies?

We have added the following caveat:

“This of course assumes highly ordered superlattices with large mean free paths, but large mean free paths in these materials should be in general be facilitated by the large a .”

6. Significance of the scattering data. The authors have made a great effort in the automatic data analysis, part of which is present in Fig. S3. However, a direct visual of the scattering function in linear scale in the 0.5 – 3 meV range is important and will make the existence of the peaks e.g. in Fig. 4B more clear.

We have added plots of the raw scattering data, $S(E)$, as well as plots of $S(E)/E$ into the SI. We agree that this is very useful to include for several reasons: (i) it highlights the difficulty in the analysis, (ii) the features we identify through the analysis are visible (upon very close inspection!) in the raw data for most samples, and (iii) Although the features are very small, when they are visible, they are visible on both the neutron energy gain and energy loss sides of the spectrum.

7. Figure S3D suggest a weakness in the modelling above 4 meV, which is unlikely to affect

conclusions but needs to be addressed. In the Supplementary Material, the authors state that the TA Van Hove singularity is at 6 meV. This is inconsistent with the core phonon term being shown as linear in $g(\omega)/\omega$.

Our rationale for fitting the core phonon term as $\propto \omega^2$ comes from the Debye model for the acoustic branch in 3D going as $g(\omega) = \frac{\omega^2}{2\pi^2 v_g^3} \rightarrow \frac{g(\omega)}{\omega} \propto \omega$ for $\omega < 6\text{meV}$.

Minor points:

- Line 26: NC is not defined.

This has been corrected

- Line 35: body-centered tetragonal and rhombohedral assemblies of cubic/cuboctahedron particles have been observed as well, and a phase diagram has been derived as a function of shape and truncation (see Wetterskog et al., Nanoscale, 2016, DOI: 10.1039/C6NR03776C and references therein). Where are the assemblies herein poised in such a phase diagram?

This is a nice work, with careful characterization of the superlattices formed with Iron Oxide NCs. The use of magnetic NCs and application of a field is particularly intriguing! We have added this as a reference in the introduction. These materials use NCs comparatively large compared to those used in our work, and are presumable capped with the native oleate ligands, (long carbon backbone -> weak spring), used in the synthesis. Superlattice phonons will occur at low energies (~3-4 times lower than the 3.3nm/DDT NCs).

This is good time to mention an important point: Using INS, materials with $g_{SL}(\omega)$ features below ~0.3 meV or so (such as that in the manuscript above) will be hard to characterize, due to the strong overlap of the scattering from $g_{SL}(\omega)$ with the QENS signal and resolution function of the instrument. So although they should be present purely from a 'degrees of freedom' argument, they will be difficult to see!

- Line 41: most, but not all vibrational modes of the organic ligands are at higher energies.

Good point, for the thiol ligands, longer ranged vibrations of the carbon backbone occur at lower energies (~7meV+, see for example 10.1038/nature16977). We have updated the line to read:

“**A majority of** the vibrational modes of organic ligand species, $g_{Lig}(\omega)$ occur at high energies (i.e. ~100 meV and above).”

- Line 80: the use of the indices ij,k, is not understandable here without further definition.

We have added a definition for the ij. The k subscript was not an index, but rather to specify the parameter belonged to the spring constant k. It was not necessary, and has been removed to avoid confusion.

- Line 87: TA/LA peak energies should be referred to also as Van Hove singularity.

We have modified the sentence to read “we can expect TA/LA **Van Hove singularities** (peaks in the $g_{SL}(\omega)$) at energies”

- Line 95: 3.27 meV.

Thanks for spotting this. We have corrected this!

- Line 150: QENS rather than QUENS.

Corrected.

- Line 152-154: revise grammar. 'do' hurts the readability and the subject of 'reduce' is unclear.

We have reworked the sentence.

- Line 368: specify the temperature for each measurement in the caption.

We have added this to the caption.

- Line 382: the information of Fig. 4C is insufficiently discussed in the main manuscript.

In 4C, we plot the integrated QENS signal as a function of temperature for one of our samples. QENS is a very powerful technique, that can and has been used to systematically study different diffusional motions, including reorientations, which are expected. Here, we merely use the QENS data to illustrate that at around $\sim 175\text{K}$, anharmonic, reorientational motion of our ligands freeze out. This helps to understand our INS data, as well as recent observation of structural changes in PbS NC-solids using small angle x-ray scattering (10.1021/acsnano.8b01643).

- For SI Fig. 2: specify what type of absorption spectroscopy was used and with what experimental setup.

We have added this information to the SI.

Reviewer #3 (Remarks to the Author):

The manuscript looks at super-lattices made of PbS nanocrystal colloids connected together through various ligands. The mechanical properties of these structures are tuned by changing the number of atoms in the colloids (hence their mass) and by varying the length of the ligands (hence their stiffness). The manuscript argues that these super-lattices can be used as functional acoustic-metamaterials. The manuscript has several shortcomings that need to be adequately addressed before I can recommend publication in Nature Communications.

1) The current version of the manuscript makes it very hard to understand what is the significance of this work in the context of metamaterials. Very little evidence is given of what kind of functional metamaterials could be engineered with this technique. For instance (and this is just an example), a hot topic of research currently is how to implement topological metamaterials that support unidirectional surface modes (Kozin et al, Phys Rev B 98, 125115, 2018). This functionality relies on the existence of Dirac cones in the phononic crystal (Torrent and Sanchez-Dehesa, Phys Rev Lett 108, 174301, 2012). Can the NC super-lattices be designed to have this desirable feature? Another example alluded to in the introduction is isolation. There are several methods to achieve acoustic isolation with passive (Mei et al, Nat Comms 3, 756, 2019) and active (Popa et al, Nat Comms 9, 5299, 2018) metamaterials. Where would the structure presented in the manuscript fit in the space of existing methods?

and

2) How is the NC lattice different from what has been done before given that structures made of PbS NCs connected through various ligands have been reported before (see [8]). Essentially, what exactly is the novelty brought by this work?

We have revised the abstract, introduction, and discussion to make it clearer to a reader what the novelty of the manuscript is.

In short, our paper presents the first experimental data proving the existence of superlattice-phonons in assemblies of colloidal nanocrystals and demonstrating that these phonon modes can be systematically engineered. This is a proof-of-principle experiment using simple structures; however, we agree with reviewer 1 who says, “the work will attract a lot of attention and be used as stepping stone for the next generation of experiment(s).” There is a vast amount of literature dealing with the growth of NCs, control and design of their ligands, and the formation of NC-superlattices. This large body of existing literature can be readily exploited in the design of more complex phonon-engineered NC-superlattices.

In the discussion, we now explain this potential:

“The design of specialized organic⁴² and inorganic⁴³ ligands can enable more complex inter-NC mechanical interactions, beyond the stretch/compressive forces provided by the thiol ligands investigated here. Including the full range of elastic interactions, such as sheer and torsional forces, would unleash the power of complex phonon design algorithms.⁴⁴ Furthermore, by controlling the shape, and therefore exposed surfaces of the NCs, the functional group of the ligands can be tailored to link between specific NC facets,^{7,8,45} enabling the definition of specific directional interactions in the superlattice. Although assembly techniques for NC-superlattices must continue to be developed so that they can compete with macroscopic metamaterials in terms of complexity; NC-superlattices open up a large parameter

space in design as well as in the resulting length and energy scales which are not easily achievable with existing platforms.”

3) The manuscript shows that various components of the lattice are tunable (e.g. Fig. 2), but what is the space of effective stiffness and NC mass that is available in these structures, and how accurately can we design for a given stiffness or NC colloid mass?

This is a good question. The range of masses can be determined by taking into account typical NCs sizes and densities (i.e., composition). This would be from roughly 10^{-24} kg to 10^{-18} kg. We are preparing a future manuscript looking a range of effective stiffnesses that is achievable as the calculation complexity open by the design space is beyond the scope of an initial proof-of-concept manuscript. As we explain in the paragraph cited above that has been added to our discussion, there is a wide range of available organic or inorganic moieties. Furthermore, the effective properties depend on how they are bound to the NC surfaces and with what density, etc. Again, one can assume effective spring constants that range across several orders of magnitude (e.g., 1 N/m to 10^5 N/m).

4) Figs. 2D, 3A, and 3B are hard to interpret without numbers on vertical axis. Is the g_{SL} peak the only interesting feature in these plots?

The two broad peaks of the $g_{SL}(\omega)$ are the key features. At higher energies, small amounts of noise in the measured signal gets amplified by the thermal factor when converting from the measured scattering intensity to $g(\omega)$. This leads to what may look like sharp “features”, however any such ‘features’ above 2 meV have errors larger than the computed $g_{SL}(\omega)$. We have modified all plots in the main text to go from 0 meV to 2 meV, to avoid any confusion caused by the noise at higher energies.

Also, it would be interesting to know how the expected and measured g_{SL} compare.

This is why we performed the theoretical calculations on the disordered mass spring systems. We now state this explicitly:

“For PbS NC superlattices studied here, which will be disordered due to their preparation through large scale solution processing, we can expect a $g_{SL}(\omega)$ between the two extremes of Figure 2D.”

We have included sections headings in the Results section to guide a reader better through the theory and experiment.

5) The stiffness shown in the caption of Fig. 2 has no units. The term NC is never defined.

Both have been added.

Reviewers' Comments:

Reviewer #1:

Remarks to the Author:

The authors have answered all of my questions satisfactorily. I have no further comments. The answers to the points raised by the other referees seem convincing as well.

Reviewer #2:

Remarks to the Author:

The revised manuscript presented by the authors satisfactorily addressed the raised criticism and is in my opinion acceptable for publication.

Minor point: in the main manuscript, Fig. 1 with schematic depiction of the scattering contributions, 'ect.' should be 'etc.'.

Raphael Hermann

Reviewer #3:

Remarks to the Author:

The revised manuscript addresses adequately all the issues raised in my previous report except for the first and third issues. It is still not clear to me what is the relevance of this work in the context of acoustic metamaterials. To answer this issue, the revision adds some vague statements such as "NC-superlattices open up a large parameter space in the design (...) which are not easily achievable with other platforms". What are these parameters difficult to achieve with other methods? What is the large space of material parameters mentioned by the authors? Specific examples would support much better the argument that the proposed lattices are suitable for functional acoustic metamaterials. For instance, based on the insight gathered from the experimental result, what composition would provide a Young's modulus, shear modulus, and/or mass density "unreachable by traditional architecture"? One illustrative example would suffice.

Furthermore, the applications promised by the abstract ("energy transport, harvesting, or isolation applications", "the creation of novel phonon-based devices, including photoacoustic systems and phonon-communication networks") look very exciting but the authors should clearly spell out how the presented work enables these applications.

The authors' answer to my third comment is not reflected in the revision. Also, it is not clear to me how the authors calculated the range of k listed in their answer (1N/m to 10^5 N/m). Fig. 2 shows $k < 100$ N/m for the considered super-lattice. What kind of lattice composition would provide the higher limit?

In my opinion, the manuscript is still not suitable for publication in its current form. Since better metamaterials is the potential application of the super-lattices proposed in this work, the authors should discuss the enabled effective material parameters needed by the metamaterials community. Such a discussion would strengthen the paper significantly.

Response to Reviewers

Reviewer #1 (Remarks to the Author):

The authors have answered all of my questions satisfactorily. I have no further comments. The answers to the points raised by the other referees seem convincing as well.

Thank you for the positive assessment!

Reviewer #2 (Remarks to the Author):

The revised manuscript presented by the authors satisfactorily addressed the raised criticism and is in my opinion acceptable for publication.

Minor point: in the main manuscript, Fig. 1 with schematic depiction of the scattering contributions, 'ect.' should be 'etc.'.

Thank you for the positive assessment and spotting this typo! It is corrected.

Reviewer #3 (Remarks to the Author):

The revised manuscript addresses adequately all the issues raised in my previous report except for the first and third issues.

...The revision adds some vague statements such as "NC-superlattices open up a large parameter space in the design (...) which are not easily achievable with other platforms". What are these parameters difficult to achieve with other methods? What is the large space of material parameters mentioned by the authors?

We apologize that this sentence was not as precise as it could be. We are referring to the length and energetic scale of the superlattice phonons as the novel component. NC superlattices offer the potential to engineer phonons on the nanometer scale, instead of at the micron or centimeter scale. NC superlattices offer phonons at the 100s of GHz to THz frequency range, not in the microwave to GHz range. Our sentence now reads:

“Although assembly techniques for NC-superlattices must continue to be developed so that they can compete with macroscopic metamaterials in terms of complexity, NC-superlattices enable phonon engineering at length and energy scales not easily achievable with existing platforms.”

Specific examples would support much better the argument that the proposed lattices are suitable for functional acoustic metamaterials. For instance, based on the insight gathered from the experimental result, what composition would provide a Young's modulus, shear modulus, and/or mass density "unreachable by traditional architecture"? One illustrative example would suffice.

The entire point of our paper is not a perspective article outlining all applications of nanocrystal superlattice phonons. We rather introduce the concept of nanocrystal superlattice phonons and

provide the first experimental demonstration of their existence. We hope our manuscript inspired the community to go out and make some of the examples you wish to see!

By functional metamaterials, we are also discussing materials that also have not only phononic properties, but also optical or electronic functionality. Our paper focuses on a material set (PbS NC superlattices), which are already widely employed in opto-electronic applications.

We have revised the paragraph highlighting the novelty of independent tunability of phononic and electronic properties enabled by these systems:

“Moreover, the ability to independently tune phononic and electronic properties in NC-superlattices is particularly promising. Phonon engineering at the atomistic level is typically limited by the fact that electronic and phononic properties are intrinsically linked. In the case of NC superlattices, electronic properties (like phononic properties) are tunable by the choice of NC size,^{46,47} shape,⁴⁸ chemical composition,^{49,50} and ligand⁵¹. However, electronic coupling between neighboring NCs and the reorganizational energy associated charge transfer govern electronic properties, while the mechanical properties of the ligand and the mass of the nanocrystal dominate the long range superlattice vibrations. Thus, electronic properties could be changed while phononic properties remain the same by, for example, placing a shell around the NC cores that changes the NC bandgap and electronic coupling between neighboring nanocrystals but keeps the NC masses the same. Alternatively, switching the ligand (e.g., from ethanedithiol to benzenedithiol) could change the effective spring constant of the system while keeping the electronic properties the same (i.e., inter-NC spacing and electronic coupling would be similar). Furthermore, the ability to tune the spatial confinement of the charge carriers (e.g., with multilayer shells, alloys, and electronegativity of ligands), to select optical band gap (e.g., with NC size), and to introduce phononic bandgaps (e.g., with complex superlattice structures using NCs of different masses), provides opportunities to selectively tune electron-phonon, phonon-photon, and phonon-phonon interactions in NC superlattices.”

The authors' answer to my third comment is not reflected in the revision.

We have another manuscript in preparation that goes into full computational detail for different types of ligand systems. This is beyond the scope of the present text, where we are working on providing the first experimental demonstration of the existence of superlattice phonons. A short looking into molecular-mechanics should suffice to accept that molecular engineering provides a rich parameter space with which to tune interactions. This is why we write:

“Phonon engineering has been predominately demonstrated with acoustic metamaterials^{18–20,38,39} on a macroscopic scale.^{40,41} The design of specialized organic⁴² and inorganic⁴³ ligands can enable more complex inter-NC mechanical interactions, beyond the stretch/compressive forces provided by the thiol ligands investigated here. Including the full range of elastic interactions, such as sheer and torsional forces, would unleash the power of complex phonon design algorithms.⁴⁴ Furthermore, by controlling the shape, and therefore exposed surfaces of the NCs, the functional group of the ligands can be tailored to link between specific NC facets,^{7,8,45} enabling the definition of specific directional interactions in the superlattice.”

Also, it is not clear to me how the authors calculated the range of k listed in their answer (1N/m to 10^5 N/m). Fig. 2 shows $k < 100$ N/m for the considered super-lattice. What kind of lattice composition would provide the higher limit?

Subscripts are important. The effective spring constant describing the interaction between two NCs is $k = N_{lig} * k_{lig}$. As we discuss, depending on the NC size and superlattice structure, N_{lig} can actually be quite large. 10^3 is not unreasonable for larger ($r \sim 4$ nm NCs with face sharing nearest neighbors). If k_{lig} is 10-100, then $k \sim 10^4 - 10^5$ N/m.

In my opinion, the manuscript is still not suitable for publication in its current form. Since better metamaterials is the potential application of the super-lattices proposed in this work, the authors should discuss the enabled effective material parameters needed by the metamaterials community. Such a discussion would strengthen the paper significantly.

We are not claiming to be able to do what is being done/has been done by the metamaterials community better with this system. Rather, nanocrystal superlattices opens up metamaterials design at new energy and length scales, and in materials where additional functionalities (e.g., tunable optical and electronic properties) are well established.